# ICDA: Interactive Causal Discovery through Large Language Model Agents

## Abstract

Large language models (**LLMs**) have emerged as a powerful method for causal discovery. Instead of utilizing numerical observational data, LLMs utilize associated variable *semantic metadata* to predict causal relationships. Simultaneously, LLMs demonstrate impressive abilities to act as black-box optimizers when given an objective $f$ and sequence of trials. We study LLMs at the intersection of these two capabilities by applying LLMs to the task of *interactive causal discovery*: given a budget of $I$ edge interventions over $R$ rounds, minimize the distance between the ground truth causal graph $G^*$ and the predicted graph $\hat{G}_R$ at the end of the $R$-th round. We propose an LLM-based pipeline incorporating two key components: 1) an LLM uncertainty-driven method for edge intervention selection 2) a local graph update strategy utilizing binary feedback from interventions to improve predictions for non-intervened neighboring edges. Experiments on eight different real-world graphs show our approach significantly outperforms a random selection baseline: at times by up to 0.5 absolute F1 score. Further we conduct a rigorous series of ablations dissecting the impact of each component of the pipeline. Finally, to assess the impact of memorization, we apply our interactive causal discovery strategy to a complex, new (as of July 2024) causal graph on protein transcription factors. Overall, our results show LLM driven uncertainty based edge selection with local updates performs strongly and robustly across a diverse set of real-world graphs.

## 1 Introduction

Given a set of variables $X_1, ..., X_n$, the *causal discovery* task involves finding a directed causal graph $G^*$ on the nodes $X_1, ..., X_n$ whose edges capture causal relationships between the *parent* (source) and *child* (target). Often, observational data can be collected for the variables $X_1, ..., X_n$. This data can then be used to predict an initial causal graph $G_0$ using statistical causal discovery techniques (Spirtes & Zhang, 2016). Recently, Large language models (**LLMs**) have emerged as a competitive alternative method for predicting causal graphs (Kıcıman et al., 2024; Abdulaal et al., 2024; Chen et al., 2024). Unlike pre-existing statistical methods, LLMs require no observational data (Kıcıman et al., 2024), instead relying purely on semantic metadata such as variable names and descriptions. Another line a work (Yang et al., 2024) investigates the abilities of LLMs to act as in-context black-box optimizers. Given an objective function $f$ and an evaluation budget $B$, the LLM is tasked with finding a maximizer $x^*$ of $f$ by sequentially proposing queries $\{x_i\}_{i=1}^{B}$ and observing their associated values $\{f(x_i)\}_{i=1}^{B}$. Taken together, these directions suggest a powerful new application of LLMs: *interactive causal discovery*.

Given an initial predicted causal graph $\hat{G}_0$ and a series of intervention rounds $1, ..., R$, the interactive causal discovery problem involves minimizing the distance $d(\hat{G}_k, G^*)$ between the predicted causal graph $\hat{G}_k$ at round $k$ and the true causal graph $G^*$ through a sequence of targeted interventions on edges. This requires the LLM to solve two key sub-tasks:

1. **Intervention selection**: Selecting which edges $(X_i, X_j)$ to intervene in the next round.

2. **Graph updates**: Updating the predicted causal graph from $\hat{G}_{k-1}$ to $\hat{G}_k$ given binary feedback based on the outcome of the previous interventions.

We propose to solve this task with the Interactive Causal Discovery Agent (**ICDA**): a novel LLM agent uncertainty-driven approach. Uncertainty estimates are predicted and maintained for each unknown edge $e \in \hat{G}_k$. Edges are then selected for intervention by prioritizing those with the highest uncertainty. When feedback is received on the selected interventions, pairwise-local updates on both edge predictions and uncertainty estimates are performed for each edge sharing a parent or child variable with an intervened edge. This process continues for $R$ rounds with $I$ edges selected for intervention each round. We benchmark ICDA on eight real world causal graphs, finding uncertainty driven selection with local updates far outperforms a baselines. In summary, we make the following contributions:

- The interactive causal discovery problem as a novel application of LLM capabilities.
- LLM based uncertainty guided intervention selection as a policy for prioritizing which edges to intervene on.
- A local update strategy for robustly updating the predicted graph $G_k$ with binary intervention feedback.
- Ablations rigorously evaluating the contribution of each pipeline component and other discovery strategies.

## 2 BACKGROUND AND RELATED WORK

**Causal Discovery and LLMs** The causal discovery task aims to learn causal relationships from observed empirical data (Peters et al., 2017; Spirtes & Zhang, 2016). Many proposed algorithms exist (Spirtes et al., 1993; Yu et al., 2019; Nauta et al., 2019; Zheng et al., 2018; Chickering, 2002) attempting to solve the causal discovery problem. However, these methods are known to struggle on real world graphs where observations are noisy or common structural assumptions are violated (Chevalley et al., 2023; Tu et al., 2019). Recently, LLMs have emerged as an alternative approach to causal discovery (Kıcıman et al., 2024; Abdulaal et al., 2024; Vashishtha et al., 2023; Li et al., 2024; Lampinen et al., 2023). Kıcıman et al. (2024) first investigated the capability of LLMs to act as zero-shot causal discovery agents using only semantic information and pairwise prompting on each variable pair. Follow-up work (Abdulaal et al., 2024) further improves LLM predictions with observational data by selecting for predictions which maximize data likelihood. Vashishtha et al. (2023) utilize *triplet prompting* to prevent cycles when the causal graph is acyclic. They show only a topological ordering on variables is required for many common causal reasoning tasks (Chu et al., 2023). Other works (Zhou et al., 2024; Chen et al., 2024) benchmark LLMs across a range of causality related tasks including causal discovery and causal inference confirming that LLMs struggle with integrating numerical data.

Another line of work more related to our proposed interactive causal discovery problem studies how to incorporate background knowledge into causal discovery algorithms (Meek, 2013). Define a set of *background knowledge* as the tuple $\mathcal{K} = (F, R)$, where $F$ specifies a set of "forbidden" graph edges and $R$ specifies a set of "required" graph edges. Meek (2013) presents an algorithm for constructing a causal graph consistent with $\mathcal{K}$ by leveraging an assumed structural DAG (directed-acyclic) property. Building on Meek (2013), Chickering (2002) proposes a greedy search algorithm that performs well in practice. In contrast to these works, our proposed algorithm utilizes LLMs to reason about the semantic/physical, as opposed to formal/structural, relationships between variables and edges in causal graphs. For this reason we are not required to make any DAG like structural assumptions common in the causal discovery literature. This is desirable as in practice many real-world causal graphs are cyclic and poorly structured (Zhu et al., 2024; Huang et al., 2021).

**LLMs as Optimizers** Another growing line of work utilizes LLMs as black-box optimizers (Yang et al., 2024; Roohani et al., 2024). Yang et al. (2024) introduce the notion of an LLM as a generic optimizer and use it to optimize performance objectives stemming from a range of tasks including linear regression and mathematical word problems (Cobbe et al., 2021). Other works (Madaan et al., 2023; Havrilla et al., 2024) examine the self-refinement capabilities of LLMs where the LLM must reason and self-improve on earlier responses. A growing number of papers apply LLMs to optimal experiment design and discovery (Roohani et al., 2024; AI4Science & Quantum, 2023; Gao et al., 2024; Majumder et al., 2024; Jansen et al., 2024). Roohani et al. (2024) apply LLMs to gene discovery tasks which aim to find highly-influential parent genes affecting the regulation of

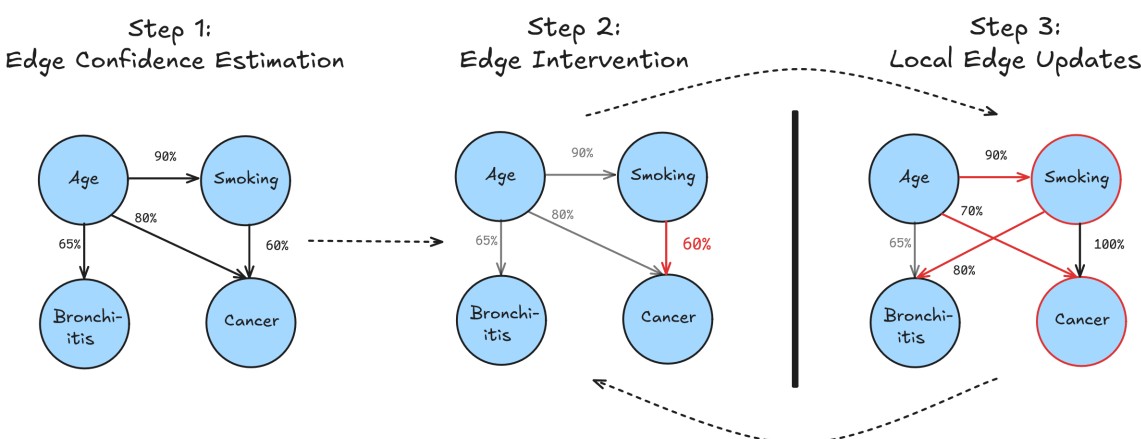

Figure 1: Diagram of the interactive causal discovery process through LLMs. The process begins by predicting edges and confidences for each edge. Interactive discovery then proceeds by selecting the most uncertain edges for intervention. The LLM then updates its predictions and confidences for edges adjacent to the intervened edge. Note: only edges predicted as present are shown.

a downstream target gene. Majumder et al. (2024); Jansen et al. (2024) both present benchmarks evaluating the ability of LLMs to perform real-world and synthesized discovery tasks.

## 3 METHOD

**Setup** As input we are given a set of variables $X_1, ..., X_n$ with associated metadata including variable names and variable descriptions. We define the notation $Y \to X$ to indicate when $Y$ is a causal parent of $X$ and the set of causal parents of a variable $X$ as $Pa(X) = \{X_i : X_i \to X\}$. We can then consider the directed ground truth causal graph $G^* = \{(X_i, X_j) : X_i \in Pa(X_j)\}$ with unlabeled and unweighted edges. Note: The only assumed graph structure is simplicity i.e. no self-edges or multi-edges. No additional structure on the graph (such as acyclicity) is assumed. We can frame the prediction of $G^*$ as an edge-wise binary classification problem over the complete graph $K_n$, where an edge $(X_i, X_j)$ has the label $l_{ij} = 1$ if $X_i \to X_j$ and $l_{ij} = 0$ otherwise. $G^*$ can then be written as a collection of ground truth labelings $G^* = \{(X_i, X_j, l_{ij}) : 1 \le i \ne j \le n\}$.

The *interactive causal discovery* task then aims to learn $G^*$ by interacting with the discovery environment via *interventions* on each edge $(X_i, X_j)$. We define an *intervention* on an edge $(X_i, X_j)$ as an operation revealing the ground truth label $l_{i,j}$. This intervention operation is purposefully kept abstract and could correspond to any number of real-world experimental intervention strategies including do operations (Sharma & Kiciman, 2020), conditional interventions, or instrumental variables. Interactive causal discovery then proceeds in two phases:

> **Phase 1 (Zero-shot prediction):** Produce an initial causal graph prediction $\hat{G}_0$ using available variables $X_1, ..., X_n$ plus semantic metadata.
>
> **Phase 2 (Interactive Discovery):** Over a series of $R$ rounds, propose $I$ edge interventions on $(X_i, X_j)$ each round and receive binary feedback on $l_{ij}$. Use this to produce an updated prediction $\hat{G}_{r-1} \to \hat{G}_r$

We evaluate the accuracy of a prediction $\hat{G}$ using the F1 objective, i.e.

$$F1(G^*, \hat{G}) = \frac{2 \cdot \texttt{Precision}_{\hat{G}} \cdot \texttt{Recall}_{\hat{G}}}{\texttt{Precision}_{\hat{G}} + \texttt{Recall}_{\hat{G}}}$$

where $\texttt{Precision}_{\hat{G}}$ and $\texttt{Recall}_{\hat{G}}$ are computed with the label predictions $(X_i, X_j, \hat{l}_{ij}) \in \hat{G}$ and $l_{ij}$ as ground truth. The goal of the interactive discovery process is then to maximize $F1(G^*, \hat{G}_R)$.

---

**Algorithm 1** Interactive Causal Discovery Through LLMs

---

0: **procedure** LLMDISCOVERY($\hat{G}$, $R$, $I$) {

 + $\hat{G}$ is the initial causal graph prediction with confidences

 + $R$ is the number of intervention rounds

 + $I$ is the number of interventions per round}

0:  **for** $r \leftarrow 1$ to $R$ **do**

0:   # **Step 1: first select edges for intervention**

0:   $sorted\_edges \leftarrow sort(\hat{G}, key = \text{``conf''})$

0:   $interventions = sorted\_edges[: I]$ # choose $I$ most uncertain edges to intervene

0:   $binary\_feedback \leftarrow do\_interventions(interventions)$

0:   # **Step 2: update $\hat{G}$ using feedback**

0:   **for** $i \leftarrow 1$ to $I$ **do**

0:    $edge, edge\_gt \leftarrow interventions[i], binary\_feedback[i]$

0:    $adjacent\_edges \leftarrow get\_adjacent\_edges(edge)$

0:    **for** $a \leftarrow 1$ to $length(adjacent\_edges)$ **do**

0:     $\hat{G}[a][\text{``update\_confs''}] \leftarrow LLMLocalUpdate(edge, edge\_gt, a)$ # see Sec. B for the LLM prompt

0:    **end for**

0:   **end for**

0:   # average over updates for next round predictions

0:   **for** $a$ in **do**

0:    $\hat{G}[a][\text{``conf''}] \leftarrow mean(\hat{G}[a][\text{``update\_confs''}])$

0:    $\hat{G}[a][\text{``pred''}] \leftarrow \mathbf{1}_{\hat{G}[a][\text{``conf''}]>0}$

0:   **end for**

0:  **end for**

0:  **return** $\hat{G}$

0: **end procedure**=0

---

**Method** Our proposed method ICDA begins by generating a zero-shot graph prediction $\hat{G}_0$. A prediction for each variable pair $(X_i, X_j)$, $1 \leq i \neq j \leq n$, is generated by prompting an LLM to reason about $X_i \rightarrow X_j$ in a manner similar to the pairwise-prompting strategy utilized in Kıcıman et al. (2024). In addition, we prompt the LLM to reason about its confidence in the prediction and output a confidence score from 1 - 100. Section B shows the exact prompt used. To obtain a reliable confidence estimate we sample the LLM $K = 16$ times. We denote the initial confidence for $(X_i, X_j)$ as $c_{ij}^0$ and set it to be the (signed) average over $K = 16$ output confidences. The initial edge label $l_{ij}^0$ is then taken as the boolean $l_{ij}^0 = \mathbf{1}_{c_{ij}^0 \geq 0}$. This gives us the initial prediction $\hat{G}_0$.

Next, in each intervention round $r \leq R$, we sort the confidence scores $\{c_{ij}^r : 1 \leq i, j \leq n\}$ by absolute value and intervene on the $I$ edges with the lowest absolute confidence (and highest uncertainty). This reveals the ground truth labels $l_{ij}$ for for each intervened edge $(X_i, X_j)$. Using this feedback, we update the predicted edge labels for intervened edges to $l_{ij}^{r+1} = l_{ij}$ and the confidences to $c_{ij}^{r+1} = 100$. Additionally, we prompt the LLM, conditioned on the ground truth label $l_{ij}$, to update its prediction and confidence for each edge $(X_i, X_k)$ or $(X_l, X_j)$, $1 \leq k, l \leq n$ which shares a node with $(X_i, X_j)$ and has absolute confidence less than 100. We call each update to an edge $(X_l, X_k)$ a *local update*. It may be that an edge $(X_l, X_k)$ is adjacent to multiple intervened edges $(X_{i_1}, X_{j_1}), (X_{i_2}, X_{j_2})$ in a single round and thus receives multiple local updates. To manage these cases we set the next confidence $c_{lk}^{r+1}$ to the (signed) average of all individual local updates to $c_{lk}^r$. Then we set $l_{lk}^{r+1} = \mathbf{1}_{c_{lk} \geq 0}$ as before. This continues until the final round $R$ is reached.

We call the complete discovery pipeline the ICDA: Interactive Causal Discovery Agent. A diagram of the full pipeline is shown in Figure 1 and written as pseudo-code in the Appendix Section D. We report all prompts in appendix Section B.

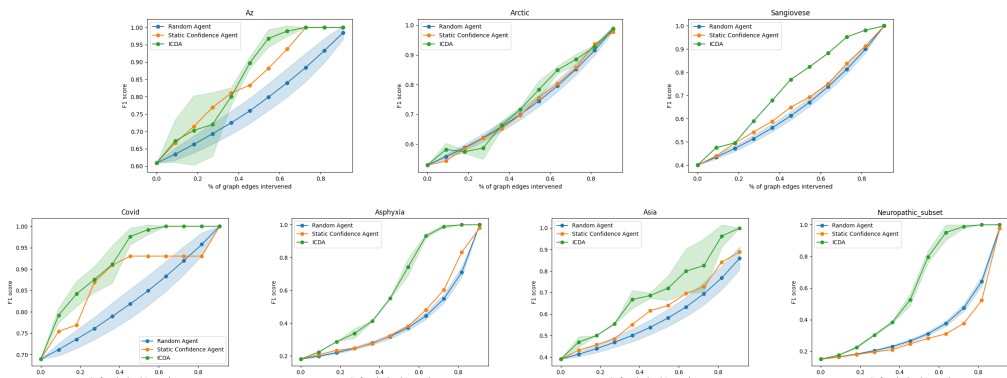

Figure 2: Results on real world graphs showing F1 score of the predicted graph against percentage of edges in the graph intervened on. ICDA almost always outperforms both the random baseline and static selection via uncertainty. Note: static confidence selection without local updates is deterministic and thus has no confidence intervals.

## 4 RESULTS

We evaluate our approach on seven real-world causal graphs. Each graph ranges from 8 - 30 variables and varies widely in causal structure (some are acyclic while others are cyclic). Details for each graph can be found in Appendix C. To produce initial zero-shot graph predictions $\hat{G}_0$ for all graphs we utilize pairwise causal prompting as in Kıcıman et al. (2024) with `Meta-Llama-3-70B-Instruct` as the base LLM. For the interactive discovery phase we then initialize all methods using $\hat{G}_0$. we compare to several baselines:

**Random selection:** Starting from $\hat{G}_0$ we randomly select edges to intervene. After receiving binary feedback we update incorrect predictions on intervened edges for the next round. We do not allow edges to be intervened twice.

**Direct LLM:** To select edges for intervention at round $r$ we directly prompt the base LLM conditioned on the entire predicted graph $\hat{G}_r$. We update edges using binary feedback by prompting the base LLM output updates conditioned on $\hat{G}_r$ and the binary feedback.

**Static confidence selection:** We select edges for intervention based on the initial confidence scores $c_{ij}$. No updates are performed beyond fixing incorrect predictions in the intervention set.

`Meta-Llama-3-70B-Instruct` is used as the base LLM when applicable. To assess performance, we plot the mean F1 score, averaged over five independent runs, against the percent of edges intervened in each graph. Results are shown in Figure 2.

**Uncertainty driven intervention selection with local updates performs best.** In all but one of the causal graphs, uncertainty driven intervention selection with the LLM utilizing interventional feedback to perform local updates performs best. Further, it outperforms the random selection baselines at nearly every round on every graph, at times by up to 0.5 absolute F1 score. The only exception to this is the Arctic sea ice graph where local updates initially perform poorly. We attribute this to the highly cyclic and thus harder-to-predict graph structure. Notably, even on graphs where the LLM proposes a poor zero-shot initial prediction, the LLM is able to recover quickly, converging to the correct causal structure with local updates. This suggests the LLM is able to effectively utilize interventional feedback even when lacking detailed domain knowledge.

**Local updates can outperform random selection even with few interventions.** Allowing the LLM to make local edge updates using intervention feedback quickly improves the predicted graph even when relatively few edges are intervened on. This behavior is particularly desirable, as in practice it may be expensive to intervene on even a small fraction of all edges. On some graphs, where

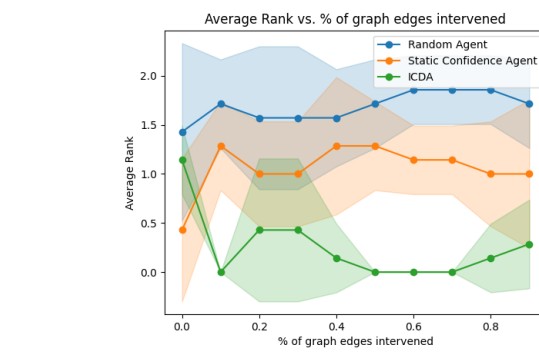

Figure 3: Average rank of each method when numbered from 0 to 2 across each timestep on each graph. The full LLM driven update agent consistently achieves rank 0 across all timesteps. Note: **lower is better**.

the initial LLM confidence estimates are good, the static confidence selection baseline without local updates is also able to quickly outperform random selection. Yet, even when the initial confidence estimates are subpar, local updates compensate and allow for the prediction to quickly improve with just a few edge interventions. This again demonstrates the broad effectiveness of local updates even when initial predictions are poor.

**Static uncertainty driven selection performs better than random selection.** Despite not fully utilizing interventional feedback, static uncertainty driven selection still outperforms the random selection baseline on five out of seven graphs. This method performs particularly well on AZ and Covid graphs where the initial LLM predictions are already reasonably good. On these graphs static uncertainty selection quickly outperforms randomly selection and is competitive even with local updates. This shows that, on a subset of the graphs, the LLM's confidence in its predictions are well-calibrated, allowing our selection policy to prevent wasting interventions on edges which are most likely already correct. However, we also see the LLM's confidence estimates can be poorly calibrated on graphs for which the initial predictions are inaccurate. See for example the Asphyxia and Neuropathic pain graphs, which start with initial F1 score less than 0.2. On these graphs the static confidence selection component struggles to outperform the random baseline.

Figure 3 aggregates the ranks of all methods across all time-steps averaged across all graphs. These results demonstrate our proposed method ICDA, combining LLM based uncertainty driven intervention selection with local updates, **significantly and consistently outperforms all baselines**. Experiments are conducted on real-world graphs with diverse causal structures establishing the practical utility of our method. In an effort to better understand the factors behind ICDA's success we conduct a number of ablations in the following section.

## 4.1 ABLATIONS

The previous section demonstrates the performance of our proposed method ICDA. In this section we ablate various components of the pipeline to understand their impact on performance.

**Impact of intervention improvements versus update improvements** As a starting point we define the *net graph improvement* in a round $r$ as the difference between the number of edges correctly classified in in $\hat{G}_r$ versus in $\hat{G}_{r-1}$. If an edge $(X_i, X_j)$ is correctly classified in $\hat{G}_r$ but not in $\hat{G}_{r-1}$ we say it has been *improved*. Recall there are two potential mechanisms of improvement for $(X_i, X_j)$: 1) $(X_i, X_j)$ was selected for intervention in the previous round $r - 1$ and feedback on the intervention was received at the start of round $r$ 2) The prediction for $(X_i, X_j)$ was updated by the LLM after receiving interventional feedback for an adjacent edge $(X_k, X_l)$. We call the former improvements *intervention improvements* and the latter *update improvements*. In a given round $r$ we are interested in how much of the net improvement for a graph is due to intervention improvements versus update improvements. To examine this, we plot both quantities in Figure 4 for the discovery

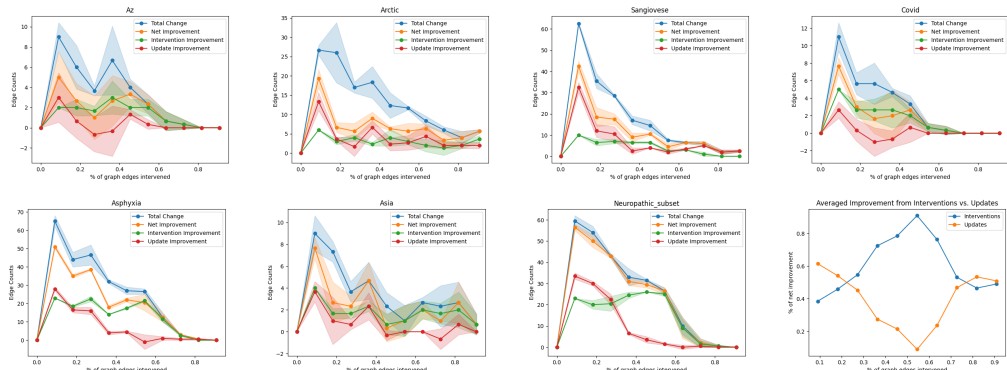

Figure 4: % Improvement from interventions vs. LLM prediction updates across timesteps. Improvement directly from LLM updates peaks early but then falls off. Improvement from interventions stays constant or improves with more interventions as confidence scores become better calibrated.

processes discussed in the previous section. In addition, we plot the net graph improvement and total number of edges changed from each round.

In all seven graphs we see both the total number of changed edges and the net improved edges peak at the first round and then decay towards zero. Notably, on some graphs there is a significant gap between net improvement and total change, indicating many edges changed during dynamic updates are misclassified after previously being correctly classified. This decline in total and net change is reflected in the number of update improvements which peak early and sharply decline to zero. This observation supports our intuition above that allowing the LLM to dynamically update edge predictions without direct intervention feedback on the edge can dramatically improve performance at small percentages of interventions. In contrast, intervention improvement accounts for a smaller percentage (less than 40%) of edge improvements early on. However, in most graphs the number of intervention improvements stays nearly constant until at least 50% of edges are already intervened. As a result, improvement from interventions grows to account for 90% of all edge improvements for rounds performed during this period. This demonstrates improvements from interventions and updates complement each other, with **update improvement driving net improvement early and intervention improvement driving net improvement later on**.

Our analysis here also confirms the effectiveness of allowing the LLM agent to update both the prediction **and** confidence for an edge. Even when only considering improvements from interventions when doing local updates, we see a major improvement over the static confidence baseline. This suggests the **updates made to edge confidence scores are equally important in achieving good performance**, allowing for sustained intervention improvement throughout the discovery process.

**Impact of Confidence Based Selection and Local Prompting**  We now ablate the impact of two key components of our discovery strategy: 1) confidence based edge selection and 2) local update prompting. To ablate 1) we directly prompt the LLM to generate a list of edges to intervene on instead of selecting via confidence. This requires us to put the entire current predicted graph $\hat{G}_r$ in-context. When dynamically updating $\hat{G}_r$ after receiving interventional feedback we remove all confidence estimates but retain the local prompting strategy. To ablate 2) we retain the same confidence edge selection proposed but replace local update prompts after with a single global update prompt containing the current prediction $\hat{G}_r$ and all recently received intervention feedback. We report the results of running the interactive discovery process with these methods in Figure 5.

We find both ablations struggle to perform better than the random baseline. Local updates without confidence selection perform well early on but fall off quickly. F1 score on the Covid graph even regresses after the initial improvements, likely due to incorrect local updates and a poor intervention selection policy. This suggests in addition to providing a strong intervention selection procedure, maintaining running confidence estimates for each edge reduces the variance of local updates from intervention feedback. Turning to the ablation for local prompting, we again find performance not

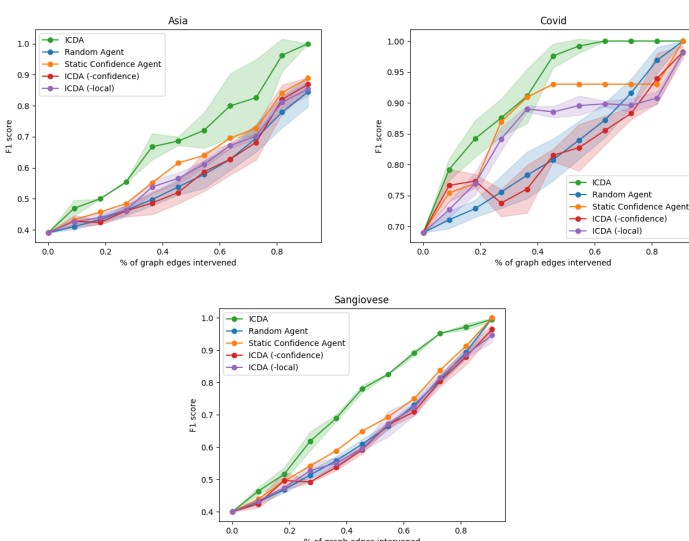

Figure 5: Ablating confidence based edge selection and local update prompting.

much better than the random baseline. Surprisingly, even on Covid where the static confidence selection performs well, confidence based selection + global updates still struggles. This indicates the base LLM is not able to correctly update the predicted graph when giving everything in context at once. This further motivate the practical importance of the local prompting procedure, which greatly simplifies the context the LLM must consider in each model call. Additionally, we notefFor large enough graphs, putting everything in context is simply not feasbile. By contrast, local prompting is easily scalable to larger graphs, albeit at a quadratic cost.

**Impact of the LLM Model Size**  The above experiments exclusively use a single base LLM ( `Meta-Llama-3-70B-Instruct`) to perform both the initial round of zero-shot edge predictions and dynamically update edge predictions/confidences using intervention feedback. Now, we examine the impact of changing both the base model size and type. In Figure 6 we initialize the discovery process with zero-shot predictions made by `Meta-Llama-3-70B-Instruct` and run local updates using the smaller `Meta-Llama-3-8B-Instruct` as well as two models from the Qwen2 series.

We find the original `Meta-Llama-3-70B-Instruct` consistently performs best on all graphs at every time step. The other 70B model, `Qwen2-72B-Instruct`, performs similarly but consistently worse. In contrast, on the Asia and Covid causal graphs, both 8B models perform worse than even the random baseline. Surprisingly `Meta-Llama-3-8B-Instruct` performs reasonably well on the Sangiovese graph, performing similarly even to the 9x larger Qwen2 70B model. Overall however these results indicate performance on the interactive causal discovery task can be substantially improved with model scale.

We next investigate the performance of different models on the initial zero-shot edge prediction task. Using the pairwise confidence estimation prompt in Section B we prompt each of four models to produce a zero-shot prediction $\hat{G}_0$ with edge confidence values. Using the predicted confidence estimates we run greedy static confidence selection procedure as in 4. Ranks for each selection procedure averaged over all graphs are plotted in Figure 7. F1 scores in each graph are reported in Figure 9 in the Appendix. As with the interactive discovery task, the smaller 8B models under perform even the random baseline. Only `Meta-Llama-3-70B-Instruct` consistently outperforms the baseline across all time steps.

**Impact of Memorization**  The success of LLMs in causal discovery stems from their immense background knowledge acquired during pre-training. This background knowledge informs the model during edge prediction and confidence calibration, allowing for strong performance even

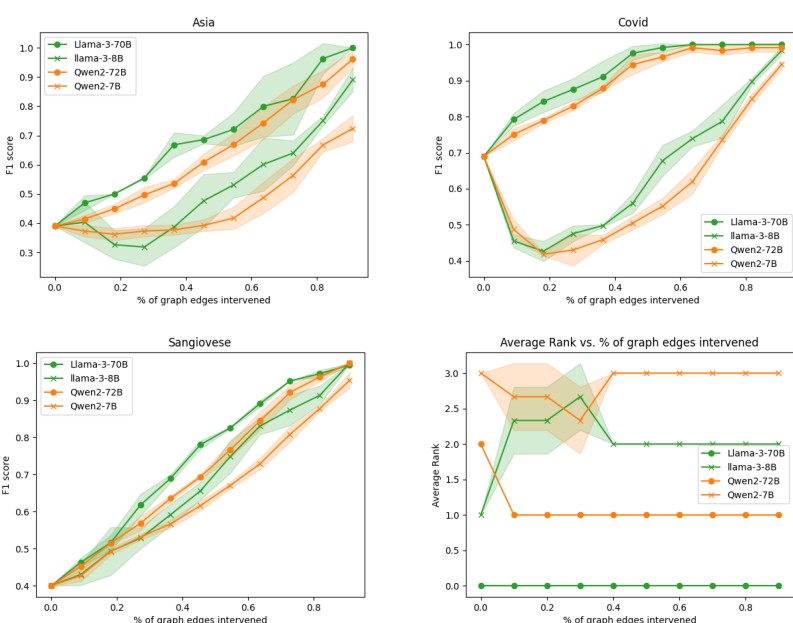

Figure 6: Performance of LLM driven interactive causal discovery on different sized models. Small LLMs (8B params) underperform the random baseline.

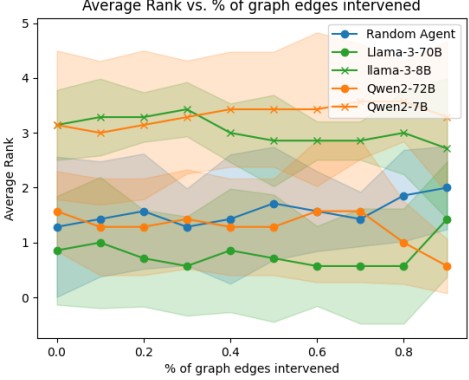

Figure 7: Static confidence based selection ranks for different models averaged across causal graphs. `Meta-Llama-3-70B-Instruct` is the only model to consistently outperform random guessing. Note: **lower is better**.

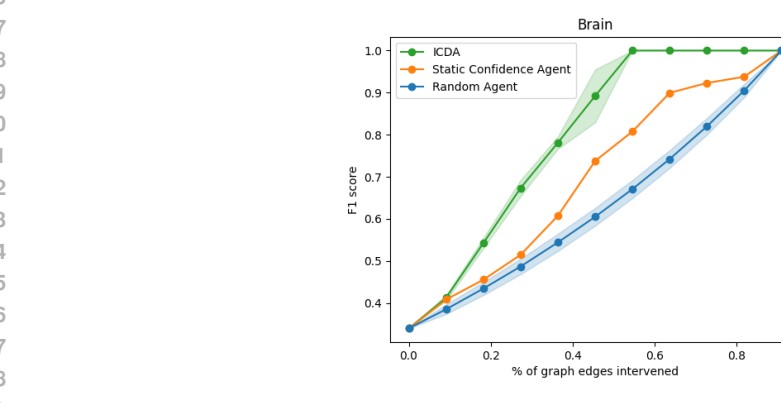

Figure 8: Performance curves of uncertainty driven selection + local prompting vs. baselines on the Brain causal graph (Zhu et al., 2024) recently published in July 2024. Both LLM driven methods perform well despite the LLM never having possibly seen the graph during training.

zero-shot. However, if benchmark graphs are contained verbatim in pre-training data, memorization becomes a significant confounding factor. To investigate to what extent memorization impacts performance we find a recently published causal graph (published in July 2024) from Zhu et al. (2024) modeling the gene regulatory network underlying 29 protein transcription factors. Because `Meta-Llama-3-70B-Instruct` finished training in 2023 this graph is guaranteed to be memorization free. Figure 8 plots the performance of uncertainty driven edge selection + local updates compared to the static selection and random baseline.

Figure 8 shows our confidence driven selection + local update approach performs very well even on graphs with minimal memorization contamination. As previously observed, local prediction updates allow for fast improvement over the random baseline even with a small number of interventions. Surprisingly, the static confidence selection approach also works well here. This indicates zero-shot edge confidence scores can be well calibrated on graphs with no contamination from memorization. We additionally note this graph has a complex causal structure with many cycles of varying lengths. This shows our method performs well even on graphs which strongly violate often assumed DAG conditions.

## 5 CONCLUSIONS AND FUTURE WORK

In this work we proposed a novel application of LLMs to interactive causal discovery. This is done by simultaneously treating the LLM as a tool for uncertainty estimation and as an optimizer utilizing interventional feedback. Our experiments confirm the proposed ICDA method significantly outperforms baselines. Further, our ablations confirm both uncertainty driven edge selection and local updates using interventional feedback as importantly contributing to the method's good performance. Future work might apply the method to larger graphs or incorporate tools relying on numerical observational data.

**Ethics Statement**  As with any work studying generative models, we note generative modeling can suffer from pre-existing biases in the training data. This behavior may help propagate existing societal biases present today.

**Reproducibility Statement**  This work utilizes only open-source models and datasets making it 100% reproducible. All benchmark graphs are documented in Appendix Section C. We additionally plan to release code in the case of an acceptance.

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

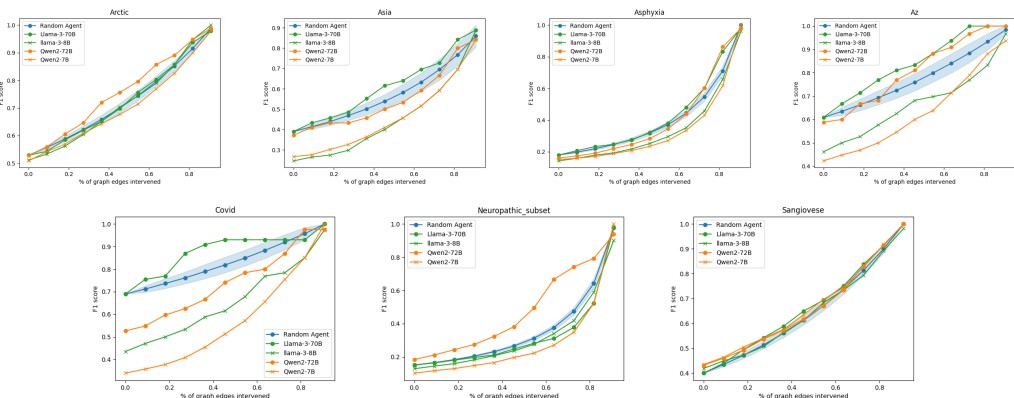

Figure 9: Static confidence selection over multiple models.

# A    STATIC CONFIDENCE SELECTION OVER MULTIPLE MODELS

# B    PROMPTS

---

**Zero-shot Confidence Estimation Prompt**

{task_description} Your goal is to understand the direct causal parents of {target}. Another variable is a direct causal parent of {target} if an intervention on the variable affects {target} and there are no other causal parents between the variable and {target}. Now, you must determine whether {parent} is a causal parent of {target}. Here is a list of all other variables to consider:
{variables_info}
Do some brainstorming, comparing relevant characteristics of both variables and then print your judgement at the end of your response enclosed in the tags ¡decision¿ YES/NO¡/decision¿. Print YES if {parent} is causal. Otherwise print NO. You should also print your confidence from a scale from 1 - 100 (with 100 being most confident) in the tags ¡confidence¿...¡/confidence¿.
Information about {target}: {target_info}
Information about {parent}: {parent_info}

---

**Parent Update Prompt**

You are a causal discovery expert. You have been given the following list of variables and tasked with predicting the true causal graph through a sequence of interventions on edges. {variables_info}
Note: each edge has an associated confidence value from 1 - 100. The presence of an edge is represented as (A-¿B,CONFIDENCE) where A is the parent and B is the child. The absence of an edge is represented as (NOT A-¿B, CONFIDENCE)
From one intervention you have discovered {intervention_feedback} Previously you predicted {intervention_prediction}
Now you should update your belief about the other edges of {parent} based on the results of the intervention. Consider the predicted edge
{other_edge_prediction}
Now you should reason about how to update your belief about the above edge based on the intervention. This means you can either keep your confidence the same, update your confidence, or change your prediction entirely. At the end of your response give your updated prediction at the end of your response in the format ¡decision¿PARENT/NOT CAUSAL¡/decision¿ ¡confidence¿CONFIDENCE¡/confidence¿. Print 'PARENT' if the edge should be present and 'NOT CAUSAL' if the edge should be absent.
You should do this in three steps.
Step 1: Brainstorm what physical causal connection there may be, if any.
Step 2: Reason about what the intervention feedback tells you. Think carefully about how similar the new child is to the intervened child.
Step 3: Give your final decision.

**Child Update Prompt**

You are a causal discovery expert. You have been given the following list of variables and tasked with predicting the true causal graph through a sequence of interventions on edges. {variables_info}
Note: each edge has an associated confidence value from 1 - 100. The presence of an edge is represented as (A-¿B,CONFIDENCE) where A is the parent and B is the child. The absence of an edge is represented as (NOT A-¿B, CONFIDENCE)
From one intervention you have discovered {intervention_feedback} Previously you predicted {intervention_prediction}
Now you should update your belief about the other edges of {child} based on the results of the intervention. Consider the predicted edge
{other_edge_prediction}
Now you should reason about how to update your belief about the above edge based on the intervention. This means you can either keep your confidence the same, update your confidence, or change your prediction entirely. At the end of your response give your updated prediction at the end of your response in the format ¡decision¿PARENT/NOT CAUSAL¡/decision¿ ¡confidence¿CONFIDENCE¡/confidence¿. Print 'PARENT' if the edge should be present and 'NOT CAUSAL' if the edge should be absent.
You should do this in three steps.
Step 1: Brainstorm what physical causal connection there may be, if any.
Step 2: Reason about what the intervention feedback tells you. Think carefully about how similar the new parent is to the intervened parent.
Step 3: Give your final decision.

## C CAUSAL GRAPHS

## D ALGORITHMS

**Algorithm 2** Interactive Causal Discovery Through LLMs

0: **procedure** LLMDISCOVERY($Xs$, $K$, $R$, $I$) {

    + Xs is a list of variable names and descriptions

    + $K$ is the number of self-consistency samples for an initial zero-shot edge prediction

    + $R$ is the number of intervention rounds

    + $I$ is the number of interventions per round}

0:    $n \leftarrow length(Xs)$

0:    $confs \leftarrow [0, ..., 0]$ {Initialize signed confidences array of length $n^2$}

0:    **for** $k \leftarrow 1$ to $K$ **do** {**First generate zero-shot prediction** $\hat{G}_0$}

0:      **for** $i \leftarrow 1$ to $n$ **do**

0:        **for** $j \leftarrow 1$ to $n$ **do**

0:          $confs[i][j] \leftarrow confs[i][j] + PairwiseConfidenceLLM(Xs[i], Xs[j])$

0:        **end for**

0:      **end for**

0:    **end for**

0:    $confs \leftarrow [False, ..., False]$ {Initialize boolean predictions array of length $n^2$}

0:    **for** $i \leftarrow 1$ to $n$ **do**

0:      **for** $j \leftarrow 1$ to $n$ **do**

0:        $confs[i][j] \leftarrow confs[i][j]/K$

0:        $preds[i][j] \leftarrow confs[i][j] > 0$

0:      **end for**

0:    **end for**

0:    **for** $r \leftarrow 1$ to $R$ **do** {**Begin interactive discovery**}

0:      $sorted\_conf\_inds \leftarrow argsort(confs)$

0:      $interventions = sorted\_conf\_inds[: I]$ {Choose $I$ most uncertain edges to intervene on.}

0:      $binary\_feedback \leftarrow do\_interventions(interventions)$

0:      **for** $i \leftarrow 1$ to $I$ **do**

0:        $edge \leftarrow interventions[i]$

0:        $edge\_gt \leftarrow binary\_feedback[i]$

0:        $adjacent\_edges \leftarrow get\_adjacent\_edges(edge)$

0:        **for** $a \leftarrow 1$ to $length(adjacent\_edges)$ **do**

0:          $confs[a] \leftarrow LocalUpdateLLM(edge, edge\_gt, adjacent\_edge[a])$

0:          $preds[a] \leftarrow confs[a] > 0$

0:        **end for**

0:      **end for**

0:    **end for**

0:    **return** $preds$

0: **end procedure**=0

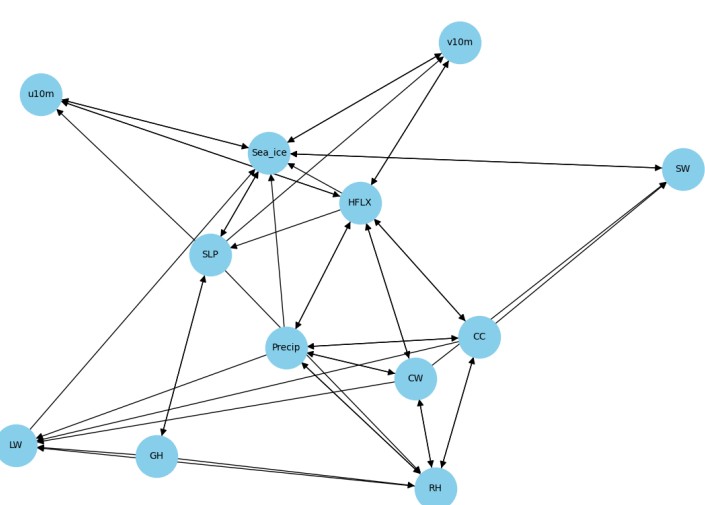

Figure 10: Arctic sea ice causal graph.

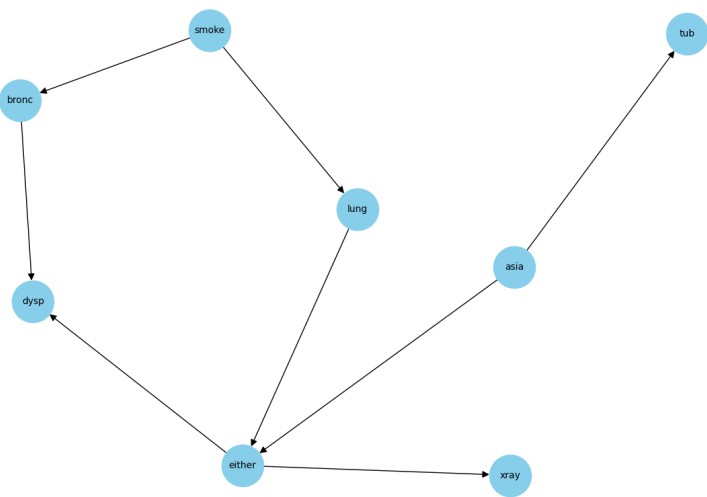

Figure 11: Asia causal graph.

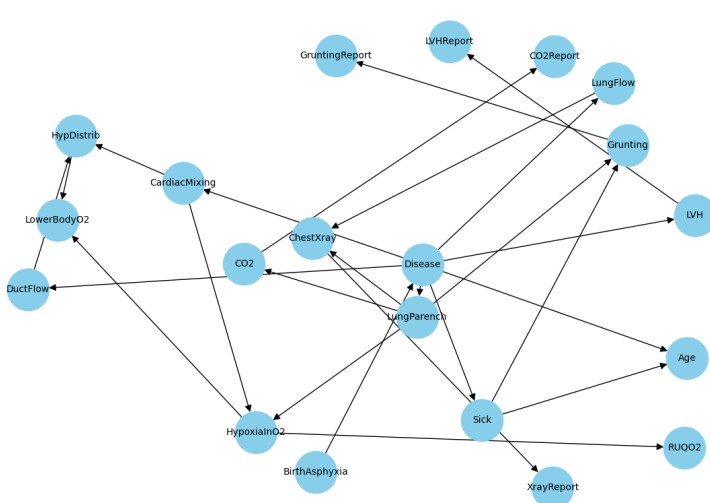

Figure 12: Asphyxia causal graph.

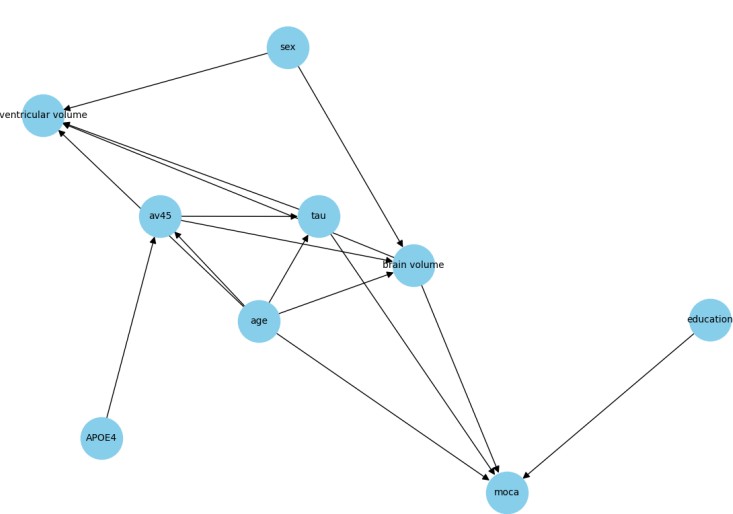

Figure 13: Alzheimers causal graph.

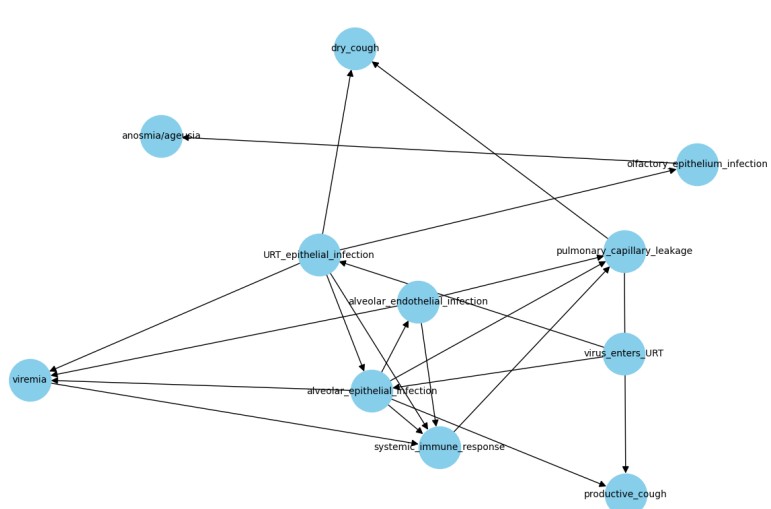

Figure 14: Covid causal graph.

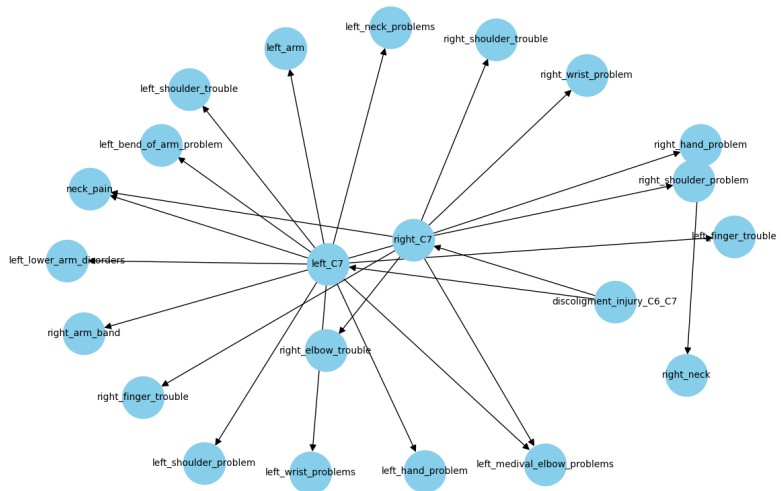

Figure 15: Neuropathic pain causal graph.

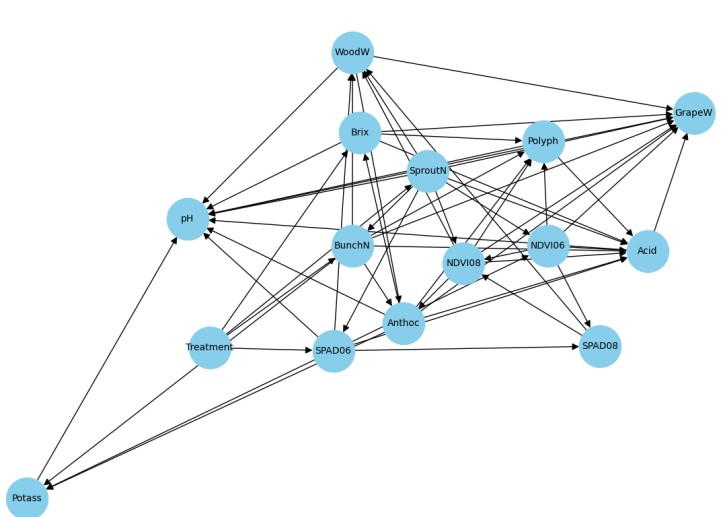

Figure 16: Sangiovese causal graph.

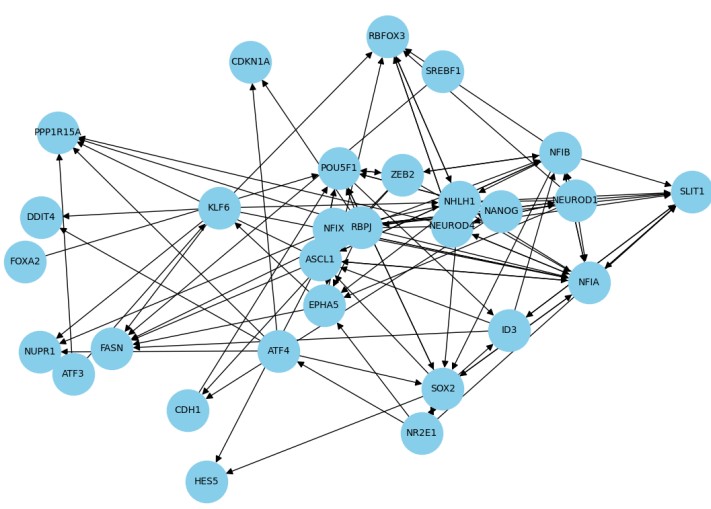

Figure 17: Brain causal graph.

