# OpenReview forum: "ICDA: Interactive Causal Discovery through Large Language Model Agents"
_ICLR.cc/2025/Conference — Submitted to ICLR 2025_

### Official Review · Reviewer_ee4g · 2024-10-20

**Soundness:** 2
**Presentation:** 3
**Contribution:** 2
**Rating:** 3
**Confidence:** 2

**Summary:**

This paper applies large language models (LLMs) to causal reasoning. Specifically, the authors prompt LLMs to address two key tasks: (1) selecting which edge to intervene on in the next round, and (2) updating the predicted causal graph. The authors demonstrate that their approach significantly outperforms a random selection baseline across eight different real-world graphs.

**Strengths:**

(1) Introducing LLMs to study causal discovery is an interesting direction.

(2) The author's writing is clear, making it easy to read and understand.

**Weaknesses:**

(1) The experimental setup is quite simple, comparing only three basic methods: random selection, direct LLM, and static confidence selection.

(2) Additionally, the comparison should include the performance of different language models, not just one.

(3) In the main experimental section, it would be better to include a table for quantitative results alongside the graphs.

(4) The experimental setup is overly simplistic, and for a conference like ICLR, the complexity of the method, theoretical analysis, and experimental thoroughness are insufficient.

**Questions:**

Refer to weakness.

---

> ### Author Response · Authors · 2024-11-22
>
> Thank you for your review! We are glad you found our direction "interesting" and paper "easy to follow". We address some of your concerns below.
>
> > The experimental setup is quite simple, comparing only three basic methods: random selection, direct LLM, and static confidence selection.
>
> We feel our analysis is fairly thorough, as we investigate in detail the performance of the following algorithms:
> - simple random edge selection
> - selection by directly promting an LLM
> - selection via direct LLM prompting + online updates to the graph via binary feedback
> - selection via static confidence
> - selection via confidence + global updates to the graph via binary feedback
> - **ICDA**: selection via confidence + local updates to the graph via binary feedback
>
> So in total we compare the performance of six different algorithms, teasing out the importance of each component used in the final method. Additionally, we note that LLMs have never been applied to this task before. As a result, there are no previous baselines relying on semantic metadata and binary edge feedback.
>
> > Additionally, the comparison should include the performance of different language models, not just one.
>
> In Figure 6 page 9 we do plot the performance of multiple models ranging from 8B - 70B parameters across two model families. These results demonstrate that smaller models struggle on this task.
>
> > In the main experimental section, it would be better to include a table for quantitative results alongside the graphs.
>
> We would be happy to include a table of results in an updated version of the paper!
>
> > The experimental setup is overly simplistic, and for a conference like ICLR, the complexity of the method, theoretical analysis, and experimental thoroughness are insufficient.
>
> Do you mind expanding on what you feel is overly simplistic about our setting, method and analysis? We are studying LLMs in a novel applicaton to interactive causal discovery and have provided an analysis which ablates each component of our method (resulting in five separate baselines). This anslysis is conducted across 8 real world causal graphs. In addition we investigate several other factors including model size/model family, and the impact of memorization on the discovery process.

---

### Official Review · Reviewer_VvSg · 2024-10-27

**Soundness:** 2
**Presentation:** 3
**Contribution:** 2
**Rating:** 5
**Confidence:** 3

**Summary:**

This paper builds on previous literature on using LLMs for causal discovery on one side, and for active black-box function optimization on the other side, to iteratively update a graph using ground-truth edges obtained through interventions.

**Strengths:**

Originalty : The submissions brings an interesting perspective by making use of the use of the literature on LLMs as optimizers.

Quality : the experiments are extensive and exhaustive, performing multiple ablation studies on the model's properties but also on aspects such as memorization.

Clarity : the paper is mostly clear in my opinion.

Signifiance : the experiments are interesting as they underline the importance of finding a subtle balance wrt local updates, between throwing the whole graph into the prompt and only modifying intervened edges.

**Weaknesses:**

Originalty/Signifiance/Quality : this point harms the correctness of the claims of the paper and is my main concern : it seems like the iterative updates do not satisfy the framework of LLMs as optimizers. From my understanding of the submission and the references, this frameworks consists in having the LLM decide on next points in the admissible space to query based on former (point, function realization) couples. But here, the next edges to query are done in a pre-determined, algorithmic manner, based on confidences, and the objective (the F1 score) is not used as an objective to optimize and is never parsed to the LLMs. The LLM is simply used on a post-hoc manner *after* having queried edges and read their associated output ground-truth labels.

Clarity : there are a few unclear points or mistakes developed in the questions.

**Questions:**

- Can you elaborate on :
a) how your algorithm satisfies the LLMs as optimizers framework? ;
b) why using the F1 metric specifically as a loss?,
c) why these specific updates on parents of intervened edges as the choice of local updates?,
d) how exactly the intervention is performed, at least in experiments?

- l.94-95 : *Building on Meek (2013), Chickering (2002) proposes a greedy search algorithm that performs well in practice.* There seems to be a confusion in time here... wrong Google Scholar citation?

- Can you increase the font of Figure 2?

- l.403-404 : *Additionally, we notefFor large enough graphs, putting everything in context is simply not feasbile*. Typo? We note that for?

---

> ### Author Response · Authors · 2024-11-22
>
> Thank you for your review! We appreciate you found our work "interesting" and our experiments "extensive and exhaustive". We address some of your comments below.
>
> > Originalty/Signifiance/Quality : this point harms the correctness of the claims of the paper and is my main concern : it seems like the iterative updates do not satisfy the framework of LLMs as optimizers... The LLM is simply used on a post-hoc manner after having queried edges and read their associated output ground-truth labels.
>
> As you say we do not directly compute the F1 score of intermediate graphs but instead receive **partial feedback** on the correctness of individual edge labels. Importantly, this partial feedback does still give some information about the F1 score of the hypothesized graph. We choose this type of feedback motivated by the experimental setting: for example biologists may be seeking to determine the causal effect of a regulatory gene R on the expression of a downstream target gene T. In practice a perturbation (knockout) can be applied to R and the resulting effect on T observed. If this knockout on R induces a statistically significant difference in the expression of T we may conclude the presence of an edge between R and T. This result could then be passed to our method and used to prioritize future experimentation.
>
> Additionally, we note the selection of edges is not pre-determined by an initial set of confidences. This is because, after recieving edge feedback, the LLM may update the confidences of adjacent edges, directly affecting the edge selection policy. In fact, one of our key findings is that **the initial set of LLM confidences become better calibrated as the LLM receives experimental feedback**. For example, the last plot in Figure 6 demonstrates the majority of graph improvement early-on comes from updates to edges made without any direct interventional feedback. Recall that edges updated due to direct interventional feedback are selected for intervention for having low-confidence scores. However, the opposite quickly becomes true as the majority of correct graph updates begin to come from useful interventional feedback as more edges are probed. This shows the updated edge confidence scores (updated using results for neighboring nodes) are the main drivers of performance later on in the discovery process, which suggests the edge confidence values are improving when compared to the static baseline (where confidences are not updated).
>
> > Can you elaborate on : a) how your algorithm satisfies the LLMs as optimizers framework? ; b) why using the F1 metric specifically as a loss?, c) why these specific updates on parents of intervened edges as the choice of local updates?, d) how exactly the intervention is performed, at least in experiments?
>
> a) The LLM as optimizer framework is defined as having an objective $f$ to maximize and a sequence of trials $x_1,...,x_N$ with associated values $f(x_1),...,f(x_N)$. The LLM must then propose a new value $x_{N+1}$ maximizing $f$ using the previous $N$ trials.
>
> In our setting, the objective $f$ as the F1 score is not directly or globally observable for the entire graph. Instead we only observe the correctness of particular edges selected for intervention by the LLM. So now each trial $x_i = \{e_{i,1},...,e_{i,I}\}$ is a set of edges selected for intervention and feedback is a set of binary labels $f(x_i) = \{l_{i,1},...,l_{i,I}\}$. As mentioned previously, we make this design choice motivated by what is often done in experimental practice. However, importantly the feedback given to the LLM can still be used to **implicitly** maximize the hypothesized graph F1 score.
>
> b) We choose F1 score simply as a way of measuring the overlap of the our hypothesized graph with the ground truth causal graph. We could choose any other overlap measure equivalently (e.g. SHD, NHD)
>
> c) To review, given binary feedback $l_{ij}$ for an edge $e_{ij}$ we update all adjacent edges $e_{ik}$ and $e_{lj}$ (so both parents and children). We chose this local strategy for two reasons:
> - We found the LLM struggled to reason about combination effects (e.g. if we put multiple sets of edge feedback and multiple edges for update in-context at once the result is significantly worse)
> - It scales better to larger graphs. Again because we do not need to put the whole graph, or large subsets, in-context.
>
> d) Because we already have access to the ground-truth graphs our intervention procedure is very simple: simply look-up the correct edge label. In practice determining this label would be done experimentally e.g. in a gene regulatory setting by introducing knockouts on the parent and observing the resulting effect on the child.
>
> >  There seems to be a confusion in time here… wrong Google Scholar citation?
>
> Thank you for catching this!
>
> > Can you increase the font of Figure 2?
>
> We plan to improve the readability of figures in an update to the paper.

---

> > ### Comment · Reviewer_VvSg · 2024-11-23
> > **Response to authors**
> >
> > Many thanks for all the clarifications. It seems that then, the optimization of the objective is supposed to be implicit. Can you provide references where this implicit optimization is done in other "LLMs for optimizers" papers? Or do you have a mathematical/biographical justification that the algorithm is equivalent to optimizing the objective directly?

---

> > > ### Author Response · Authors · 2024-11-24
> > >
> > > In real-world experimental settings there is no good way to go about measuring the F1 score of the entire predicted graph. This would require access to the ground truth graph ahead of time. Instead it is more feasible to conduct experiments testing the causal relationship between variables A and B (e.g. through gene knockouts in a gene regulatory network). Such an experiment can then be used to produce the edge-level feedback our algorithm is designed to receive.

---

> > > > ### Comment · Reviewer_VvSg · 2024-12-01
> > > > **Thanks**
> > > >
> > > > Many apologies for my delay due to personal circumstances. The paper has interesting results but I keep finding that the use of the literature on LLMs as optimizers is inappropriate in this context, making the paper significantly weaker. Thus I will keep my score. I encourage authors to explicitly drop the reference to such literature.

---

### Official Review · Reviewer_fPs7 · 2024-11-02

**Soundness:** 2
**Presentation:** 2
**Contribution:** 2
**Rating:** 3
**Confidence:** 4

**Summary:**

This work studies using LLMs to perform causal discovery in an interactive manner.  The authors propose to incorporate LLMs as an agent to produce initial graphs, and iteratively optimize the updated causal graphs by selecting proper interventions. During the selection of the intervention targets, LLMs are leveraged to provide uncertainty measures for the unknown edge. The authors show that the proposed approach can effectively outperforms simple baselines on eight real-world causal graphs.

**Strengths:**

(+) This work presents an interesting use of LLMs in causal discovery;

(+) The presentation and organization of this work are clear and easy-to-follow;

(+) Some experiments demonstrate the effectiveness of the proposed approach;

**Weaknesses:**

(-) The setting may not be realistic, since it is challenging to directly obtain the ground-truth causal edge label in each intervention;

(-) There is no guarantee that LLMs could provide valid results;

(-) Previous baselines on experimental design are neglected;

**Questions:**

1. The setting may not be realistic:
- In the proposed setting, line 144, it is assumed that one could directly obtain the ground-truth causal edge label in each intervention, which is not realistic;
- The setting significantly differs from the standard practice in the literature of experimental design [1,2,3];

2. There is no guarantee that LLMs could provide valid results:
- It is widely shown that LLMs can not provide faithful causal results [4,5], while the proposed framework heavily rely on the results of LLMs;
- Similarly, the uncertainty provided by LLMs is not warranted;

3. Previous baselines on experimental design are neglected, for example, previous works on intervention selection or experimental design [1,2,3].

Minor
- The line numbers of the algorithm are all 0;
- lots of key steps in the algorithm are not defined;

**Refereneces**

[1] Learning neural causal models with active interventions.

[2] Trust your $\nabla$: Gradient-based intervention targeting for causal discovery.

[3]  Active learning for optimal intervention design in causal models.

[4] Causal parrots: Large language models may talk causality but are not causal.

[5] Discovery of the hidden world with large language models.

---

> ### Author Response · Authors · 2024-11-22
>
> Thank you for your review! We are glad you found our paper "interesting" and that some of our experiments "demonstrate the effectiveness of our approach". We address some of your comments below.
>
> > The setting may not be realistic, since it is challenging to directly obtain the ground-truth causal edge label in each intervention; In the proposed setting, line 144, it is assumed that one could directly obtain the ground-truth causal edge label in each intervention, which is not realistic; The setting significantly differs from the standard practice in the literature of experimental design [1,2,3];
>
> To summarize, our 'edge intervention' formulation assumes the access to a noiseless operation that reveals presence/absence of a causal edge between two variables. This design is motivated by experimental settings where for example biologists may be seeking to determine the causal effect of a regulatory gene R on the expression of a downstream target gene T. In practice a perturbation (knockout) can be applied to R and the resulting effect on T observed. If this knockout on R induces a statistically significant difference in the expression of T we may conclude the presence of an edge between R and T. This can be extended to the multi-edge setting where now we examine the effect of either a single gene knockout or multiple gene knockouts simultaneously on target downstream variables. See [1] for an example of this procedure in neuronal stimuli.
>
> Further, we note that previous literature focuses exclusively on methods utilizing numerical observational and interventional data for causal discovery. In contrast, our method (and LLMs for causality in general) rely purely on pre-training, semantic metadata and experimental feedback. However, we believe both data sources (numerical and semantic) are complementary. Combining both into a single system for interactive causal discovery is promising future work.
>
> > There is no guarantee that LLMs could provide valid results; It is widely shown that LLMs can not provide faithful causal results [4,5], while the proposed framework heavily rely on the results of LLMs;
> Similarly, the uncertainty provided by LLMs is not warranted;
>
> The initial confidence estimates we get from the LLM seem to be unreliable only on some graphs but more reliable on others. For example, compare the performance of the static confidence baseline on the Covid and Asphyxia graphs. The Covid graph is relatively simple and likely highly represented in pre-training data. As a result, we speculate this is why initial confidence estimates are well-calibrated for this graph. However, the Asphyxia graph is much larger and likely less well-represented in pre-training data. As a result, the initial LLM confidences aren't much better than the random baseline.
>
> However, one of our key findings is that **the initial set of LLM confidences become better calibrated as the LLM receives experimental feedback**. For example, the last plot in Figure 6 demonstrates the majority of graph improvement early-on comes from updates to edges made without any direct interventional feedback. Recall that edges updated due to direct interventional feedback are selected for intervention for having low-confidence scores. However, the opposite quickly becomes true as the majority of correct graph updates begin to come from useful interventional feedback as more edges are probed. This shows the updated edge confidence scores (updated using results for neighboring nodes) are the main drivers of performance later on in the discovery process, which suggests the edge confidence values are improving when compared to the static baseline.
>
> > Previous baselines on experimental design are neglected, for example, previous works on intervention selection or experimental design [1,2,3].
>
> An apples-to-apples comparison with statistical methods is somewhat difficult. This is because statistical methods use numerical observational/interventional data whereas our method (and LLMs for causality in general) rely purely on pre-training, semantic metadata and experimental feedback. So the data sources each method utilizes are entirely separate and potentially complementary. We believe combining these two methods for the task of interactive discovery would be interesting future work.
>
> Additionally, for this reason we instead strive to benchmark our complete method against various weaker but natural LLM based methods (e.g. random selection, direct prompting, static confidence). We believe this benchmarking is thorough and demonstrates the necessity of our method over simply, for example, prompting a model with the entire graph.
>
> [1] Single neurons may encode simultaneous stimuli by switching between activity patterns, https://pubmed.ncbi.nlm.nih.gov/30006598/

---

> > ### Comment · Reviewer_fPs7 · 2024-11-26
> >
> > Thanks to the authors for providing a detailed explanation of my questions. Nevertheless, it seems the proposed approach is "too good to be true". It is already known that LLMs may output the "correct answers" if the inquired causal information were present during the pretraining. The use of the confidence does not have any guarantees. Hencefore, it seems quite challenging to convince the domain experts to take the proposed approach into practice use.

---

> > > ### Author Response · Authors · 2024-11-26
> > >
> > > We thank the reviewer for their response. We note that we explicitly ablate the contamination of the causal graphs we evaluate in model pre-training data by evaluating on a graph published after our model's training cutoff. Even on this graph our proposed method performs quite well. We believe this provides compelling evidence something more interesting than causal memorization is occurring. Additionally, we note that although our use of confidences has no explicit guarantees, we present detailed ablations finding that LLM confidences become better calibrated after receiving edge feedback. This provides evidence the LLM is able to act as a bayesian causal reasoner which again indicates something more interesting/useful is happening than causal memorization.

---

### Official Review · Reviewer_iLAe · 2024-11-04

**Soundness:** 3
**Presentation:** 3
**Contribution:** 2
**Rating:** 5
**Confidence:** 3

**Summary:**

The authors propose a new method for end-to-end interactive causal discovery using LLMs.
The approach comprises two main components intervention selection method based on LLM uncertainty predictions and local update strategy based on newly acquired knowledge.
The approach is based on a formulation of edge intervention. The method is evaluated on the set of 7 real-world graphs and compared against its ablations. Additional analysis is provided which covers evaluation with different LLM models and evaluation on the graph unseen during LLM training.

**Strengths:**

1. The paper is cleanly written.
2. The approach is well motivated by the literature.
3. The experimental section is extensive.

**Weaknesses:**

1. The definition of edge intervention feels unrealistic. Could the authors please provide an example of a causal operation that reveals the edge without additional knowledge or assumptions about the graph structure? It seems to me that such data might be extremely costly to obtain and the operation might in some cases be equivalent to revealing the whole graph, thus making the described approach impractical.
2. The paper lacks a discussion about the limitations of the proposed approach.

**Questions:**

1. The plots in Figures 2 and 4 are very small and hard to read.
2. In line 312 there seems to be a space missing - “weablate”
3. The citation (Sharma & Kiciman, 2020), in line 147 seems misplaced. What was the authors' intention?

---

> ### Author Response · Authors · 2024-11-22
>
> Thank you for the reivew! We are glad the reviewer found our paper "cleanly written" and our experiments "extensive". We address some of your comments below:
>
> > The definition of edge intervention feels unrealistic. Could the authors please provide an example of a causal operation that reveals the edge without additional knowledge or assumptions about the graph structure? It seems to me that such data might be extremely costly to obtain and the operation might in some cases be equivalent to revealing the whole graph, thus making the described approach impractical.
>
> To summarize, our 'edge intervention' formulation assumes the access to a noiseless operation that reveals presence/absence of a causal edge between two variables. This design is motivated by experimental settings where for example biologists may be seeking to determine the causal effect of a regulatory gene R on the expression of a downstream target gene T. In practice a perturbation (knockout) can be applied to R and the resulting effect on T observed. If this knockout on R induces a statistically significant difference in the expression of T we may conclude the presence of an edge between R and T. This can be extended to the multi-edge setting where now we examine the effect of either a single gene knockout or multiple gene knockouts simultaneously on target downstream variables. See [1] for an example of this procedure in neuronal stimuli.
>
> It's additionally worth noting that our current method does allow for the incorporation of noisy edge feedback (where now the confidence score assigned to a feedback edge may be < 100). However we did not evaluate this setting in any of our experiments.
>
> > The paper lacks a discussion about the limitations of the proposed approach.
>
> One clear limitation is that cannot make use of numerical observational/interventional data when updating our graph prediction. This is because classical statistical methods use numerical observational/interventional data whereas our method (and LLMs for causality in general) rely purely on pre-training, semantic metadata and experimental feedback. However, we believe both data sources (numerical and semantic) are complementary. Combining both into a single system for interactive causal discovery is promising future work.
>
>
> > The plots in Figures 2 and 4 are very small and hard to read.
>
> We apologize for this! We plan to improve the readability of figures in an updated version of the paper.
>
> [1] Single neurons may encode simultaneous stimuli by switching between activity patterns, https://pubmed.ncbi.nlm.nih.gov/30006598/

---

> > ### Comment · Reviewer_iLAe · 2024-11-28
> > **Thank you for the rebutal**
> >
> > I thank the authors for answering my questions.
> >
> > * **Ad.  unrealistic edge intervention setting**
> > Thank you for providing this example. Unfortunately, my concerns are not addressed. I am not an expert on biological matters but in the general setting if we perturb a variable R and observe the statistically significant change in the distribution of variable T  it is not enough to claim there is a **direct** causal relation. After all, there can be numerous variables on the path from R to T, which will all be affected. Therefore my question remains unanswered. Could the authors please provide an example of a causal operation that reveals the edge without additional knowledge or assumptions about the graph structure?
> >
> > * Question 3 has not been answered.
> >
> > Since my concerns were not addressed and no changes have been made to the manuscript I keep my score.

---

### Official Review · Reviewer_bckM · 2024-11-04

**Soundness:** 3
**Presentation:** 2
**Contribution:** 3
**Rating:** 6
**Confidence:** 3

**Summary:**

This paper proposed Interactive Causal Discovery Agent (ICDA), that uses LLMs for causal discovery through an uncertainty-driven edge intervention selection process. The method prioritizes uncertain edges for intervention and utilizes local updates from feedback, achieving strong performance on a range of real-world causal graphs. Extensive experiments validate ICDA’s robustness and adaptability, showing it outperforms zero-shot LLM prompting across diverse graph structures.

**Strengths:**

- The paper introduces a novel application of LLMs for causal discovery - using LLM defined interventions to refine causal discovery.

- ICDA is evaluated on diverse datasets including a dataset not part of the model pertaining.

- The paper is well-written and easy to follow.

**Weaknesses:**

- Lack of comparison with statistical methods.

- There is a lack of comprehensive results across models. I acknowledge the authors have presented results in Figure 6, but comparing different ICDA variants and random agents for smaller models would have been interesting.

**Questions:**

- Some of the figures done have confidence bound (last subplot for Fig 6). Am I missing something?

- What does "simplicity" mean on L138?

- Some works suggest that LLMs confidence might be unreliable, I wonder what the intuition on much better results with ICDA is in comparison to random agents?

Minor:

- Figures might benefit from increasing the font size.

- a weird indent on L254

---

> ### Author Response · Authors · 2024-11-22
>
> Thank you for your review! We are glad the reviewer found our application novel and well-presented! We address some of your comments below:
>
> > Lack of comparison with statistical methods.
>
> Statistical methods use numerical observational/interventional data whereas our method (and LLMs for causality in general) rely purely on pre-training, semantic metadata and experimental feedback. So the data sources each method utilizes are entirely separate and potentially complementary. We believe combining these two methods for the task of interactive discovery would be interesting future work.
>
> Additionally, for this reason we instead strive to benchmark our complete method against various weaker but natural LLM based methods (e.g. random selection, direct prompting, static confidence). We believe this benchmarking is thorough and demonstrates the necessity of our method over simply, for example, prompting a model with the entire graph.
>
>
> > There is a lack of comprehensive results across models. I acknowledge the authors have presented results in Figure 6, but comparing different ICDA variants and random agents for smaller models would have been interesting.
>
> We agree a more thorough investigation across models (beyond what is already done in Figure 6) would be interesting. However we instead chose to prioritize other ablations investigating the impact of different parts of the edge selection scheme and the potential impact of memorization.
>
> > Some of the figures done have confidence bound (last subplot for Fig 6). Am I missing something?
>
> Thanks for pointing this out! We can fix this in an updated version.
>
> > What does "simplicity" mean on L138?
>
> We mean to express that the graphs are simple i.e. no nodes are self-causal.
>
> > Some works suggest that LLMs confidence might be unreliable, I wonder what the intuition on much better results with ICDA is in comparison to random agents?
>
> The initial confidence estimates we get from the LLM seem to be unreliable only on some graphs but more reliable on others. For example, compare the performance of the static confidence baseline on the Covid and Asphyxia graphs. The Covid graph is relatively simple and likely highly represented in pre-training data. As a result, we speculate this is why initial confidence estimates are well-calibrated for this graph. However, the Asphyxia graph is much larger and likely less well-represented in pre-training data. As a result, the initial LLM confidences aren't much better than the random baseline.
>
> However, one of our key findings is that **the initial set of LLM confidences become better calibrated as the LLM receives experimental feedback**. For example, the last plot in Figure 6 demonstrates the majority of graph improvement early-on comes from updates to edges made without any direct interventional feedback. Recall that edges updated due to direct interventional feedback are selected for intervention for having low-confidence scores. However, the opposite quickly becomes true as the majority of correct graph updates begin to come from useful interventional feedback as more edges are probed. This shows the updated edge confidence scores (updated using results for neighboring nodes) are the main drivers of performance later on in the discovery process, which suggests the edge confidence values are improving when compared to the static baseline.

---

> > ### Comment · Reviewer_bckM · 2024-11-23
> > **Thank you for the rebuttal**
> >
> > Dear authors,
> >
> > I appreciate your efforts to answer my questions. Thank you! While most of my questions are answered, I still have a concern regarding comparison with statistical methods for causal discovery. I read your response to another reviewer suggesting that the results for statistical (w interventions) discovery optimize over different parameters, however, the way I see it is your methodology is an alternative to the statistical method, and think it would be good to compare against even simple causal discovery, PC algorithm and w intervention. Unless the authors can motivate different motivations for using both of the methods, I think it would be imp to include those.

---

> ### Author Response · Authors · 2024-11-24
>
> Thank you as well! As we mentioned above, statistical methods rely on numerical observational/interventional data for training. As a result, their performance is sensitive to the amount of numerical data available (which can vary significantly for real world problems). LLMs do rely on numerical data but instead of variable semantic metadata and pre-training. Because there is no overlap in the data used to make edge predictions it is not clear how to even go about making a fair comparison (e.g. how much numerical data should be provided to statistical methods?). Instead, the intent of our paper is to demonstrate that LLMs, as general purpose models of language, can be effectively used for interactive causal discovery provided the right agent/system design. However, we agree combining these statistical methods and LLMs would make for interesting future work.

---

### Meta-Review · Area_Chair_aGJJ · 2024-12-21

**Metareview:**

The paper introduced an approach using LLMs for causal discovery through uncertainty-driven edge intervention selection.

Strengths:

+ Studies an important problem of combining LLMs with causal discovery

Weaknesses:

+ Unrealistic assumption about edge intervention since obtaining ground truth causal edge labels directly may not be practical

+ Lack of comparison with statistical methods and previous experimental design baselines

+ Limited theoretical guarantees on LLM reliability for causal reasoning and uncertainty estimation

**Additional Comments On Reviewer Discussion:**

The reviewers are in agreement that the paper in its current form presents concerns around fundamental assumptions and methodological choices.

---

### Decision · Program_Chairs · 2025-01-22

Reject